# EFFICIENT TRAINING AND INFERENCE OF HYPERGRAPH REASONING NETWORKS

## ABSTRACT

We study the problem of hypergraph reasoning in large domains, e.g., predicting the relationship between several entities based on the input facts. We observe that in logical reasoning, logical rules (e.g., my parent's parent is my grandparent) usually apply locally (e.g., only three people are involved in a grandparent rule), and sparsely (e.g., the grandparent relationship is sparse across all pairs of people in the world). Inspired by these observations, we propose Sparse and Local Neural Logic Machines (SpaLoc), a structured neural network for hypergraph reasoning. To leverage the sparsity in hypergraph neural networks, SpaLoc represents the grounding of relationships such as *parent* and *grandparent* as sparse tensors and uses neural networks and finite-domain quantification operations to infer new facts based on the input. We further introduce a sparsification loss to regularize the number of hyperedges in intermediate layers of a SpaLoc model. To enable training on large-scale graphs such as real-world knowledge graphs, SpaLoc makes training and inference-time sub-sampling of the input graphs. To remedy the information loss in sampled sub-graphs, we propose a novel sampling and label calibration paradigm based on an information-theoretic measure *information sufficiency*. Our SpaLoc shows superior accuracy and efficiency on synthetic datasets compared with prior art and achieves state-of-the-art performance on several real-world knowledge graph reasoning benchmarks.

## 1 INTRODUCTION

Performing graph reasoning in large domains, such as predicting the relationship between two entities based on the input facts, is an important practical problem that arises in reasoning about molecular modeling, knowledge networks, and collections of objects in the physical world (Schlichtkrull et al., 2018; Veličković et al., 2020; Battaglia et al., 2016). The necessary inference rules are generally unknown and must be inferred from data, which in general are large. In this paper, we focus on learning neural networks for graph reasoning tasks. Consider the problem of learning a rule that explains the *grandparent* relationship. Given a dataset of labeled family relationship graphs, we aim to build machine-learning algorithms that can learn to predict a specific relationship (e.g., *grandparent*) based on other relationships, such as $father(x, y)$ and $mother(x, y)$.

Neural Logic Machines (NLM; Dong et al., 2019) present a method for solving graph reasoning tasks with a structured neural network. NLMs keep track of hyperedge representations for all tuples consisting of up to $B$ entities. Thus, they can infer more complex finitely-quantified logical relations than standard graph neural networks that only consider binary relationships between entities (Morris et al., 2019; Barceló et al., 2020). However, there are two disadvantages of such a dense hypergraph representation. First, the training and inference of NLMs requires simultaneously considering all entities in a domain, such as all of the $N$ people in a family relationship database. Second, they scale polynomially with respect to the number of entities considered in a single inference. Even inferring a single relation like *grandparent* requires $O(N^3)$ time and space complexity. In practice, for large graphs, these limitations make the training and inference intractable and hinder the application of NLMs in large-scale real-world domains.

To address these two challenges, we present a novel framework, called Sparse and Local Neural Logic Machines (SpaLoc), for inducing sparse relational rules from data in large domains. Our key idea is to exploit *locality* and *sparsity* in data: determining a relationship between entities

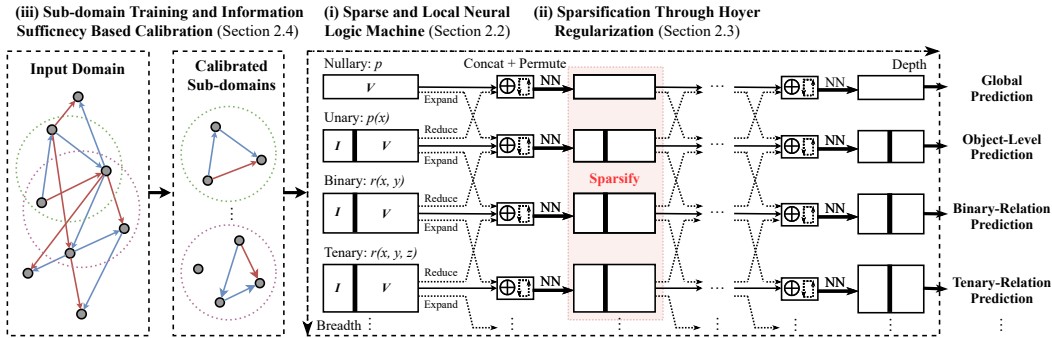

Figure 1: The overall pipeline of SpaLoc: a multi-layer neural network that applies to hypergraphs. *I and V denote the index tensor and value tensor respectively.* To facilitate training and inference in large domains, we employ a sub-graph sampling procedure and sparsification.

usually only requires consideration of a small number of additional entities, and the resulting relation usually only holds for a small number of tuples of entities. Our contribution is three-fold. First, we develop a sparse tensor-based representation for representing hyperedge relationships among entities and making inferences about them. Second, during both training and inference, SpaLoc employs a sub-graph sampling technique based on an information-theoretic measure, *information sufficiency*, which quantifies the amount of information contained in a sub-graph with respect to whether a predicate is true for some tuple of entities. Since the sub-graph sampling may violate the closed-world assumption (i.e., the information in a sub-sampled graph may be insufficient in predicting the relationship between a pair of entities), we also use the information sufficiency measure to calibrate training labels. Third, to further speed up inference on large graphs, we encourage neural networks recover *sparse* relationships among objects by using a regularization term based on graph sparsity measurements.

We evaluate SpaLoc on two benchmarks: relational reasoning in synthetic datasets (family trees and general graph reasoning) and real-world knowledge-graph reasoning. First, we show that, with our sparsity regularization, the computation complexity for inference can be reduced to the same order as the optimal complexity, which significantly outperforms the base model NLM. Second, we show that training via sub-graph sampling and label calibration enables us to learn relational rules in real-world knowledge graphs with more than 10K nodes, whereas the original NLM can be barely applied to graphs with more than 100 nodes. Finally, SpaLoc achieves state-of-the-art performance on several real-world knowledge graph reasoning benchmarks.

## 2 SPARSE AND LOCAL NEURAL LOGIC MACHINES

In this section, we develop a structured neural network that applies to hypergraphs. The fundamental structures used for both training and inference, will be hypergraphs $\mathcal{H} = (\mathcal{V}, \mathcal{E})$, where $\mathcal{V}$ is a set of vertices and $\mathcal{E}$ is a set of hyperedges. Each hyperedge $e = (x_1, x_2, \cdots, x_r)$ is an ordered tuple of $r$ elements ($r$ is called the arity of the edge), where $x_i \in \mathcal{V}$. We use $f : \mathcal{E} \to \mathcal{S}$ to denote a *hyperedge representation function*, which maps each hyperedge $e$ to a feature in domain $\mathcal{S}$. Domain $\mathcal{S}$ can be in various forms: discrete labels, numbers, and vectors. For simplicity, we describe features associated with arity-1 edges as "node features" and features associated with the whole graph as "nullary" or "global" features.

A graph-reasoning task can be formulated as follows: given $\mathcal{H}$ and the input hyperedge representation functions $f$ associated with all hyperedges in $\mathcal{E}$, such as node types and pairwise relationships (e.g., *parent*), our goal is to infer a target representation function $f'$ for one or more hyperedges, i.e. $f'(e)$ for some $e \in \mathcal{E}$, such as predicting a new relationship (e.g., *grandparent*).

### 2.1 PRELIMINARY: NEURAL LOGIC MACHINES

We provide an informal overview of the global architecture of Neural Logic Machines (NLMs); for a formal description, see the original paper by Dong et al. (2019). In contrast to typical graph neural networks that only associate representations with input edges in $\mathcal{E}$, NLMs operate on a fully-

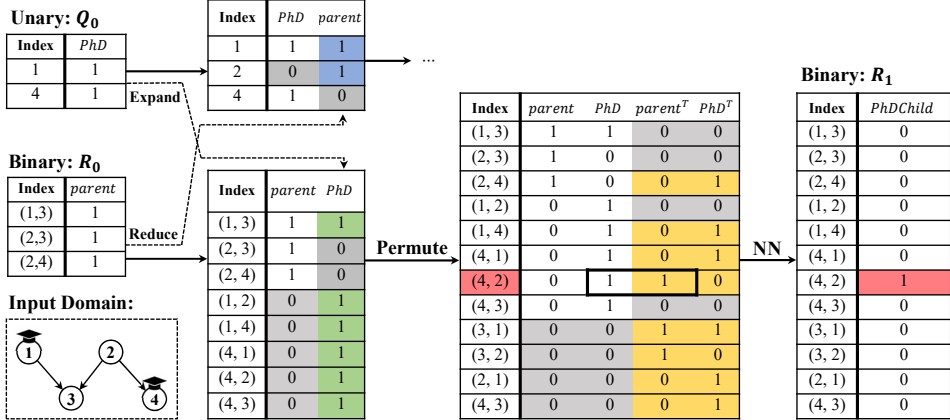

Figure 2: A running example of a single layer SpaLoc: inferring the binary relationship of *PhD-child-of* from the attribute *PhD* and the binary relationship *parent*. Blue entries denote values that are reduced from high-arity tensors, while green entries are expanded from low-arity tensors. Yellow entries are created by the "permutation" operation. Gray entries are zero paddings.

connected hypergraph. Thus, NLMs use dense tensors to represent the input, output and intermediate representations of relations on nodes, edges, and hyperedges.

For example, the input relation $parent(x, y)$ in a given hypergraph $\mathcal{H}$ can be represented as a 2D tensor $parent_{\mathcal{H}}$ of size $N^2$, where $N = |\mathcal{V}|$ is the number of entities in the hypergraph, and $parent_{\mathcal{H}}[i, j] = 1$ if and only if person $i$ is a parent of $j$. In general, a vector representation of length $D$ associated with hyperedges of arity $r$ can be represented using a tensor of size $N^r \times D$.

An NLM is a multi-layer neural network that operates on this dense hypergraph representation. At each layer, it computes a new set of features associated with the hyperedges based on the input. To fuse the information across hyperedges of different arities, NLMs use two special tensor operations, EXPAND and REDUCE, which add and reduce dimensions of the representation tensors.

**Remark**. Even when the inputs have only unary and binary relations, allowing intermediate tensor representations of higher arity to be associated with hyperedges increases the expressiveness of NLMs (Dong et al., 2019). An intuitive example is that, in order to determine the *grandparent* relationship, we need to consider all 3-tuples of entities, even though the input relations are only binary. Despite their expressiveness, NLMs based on hyperedges cannot be directly applied to large-scale graphs. For a graph with more than 10,000 nodes (e.g., Freebase (Bollacker et al., 2008)), it is almost impossible to store vector representations for all of the $N^3$ tuples of arity 3. Our key observation to improve the training and inference efficiency of NLMs is that relational rules are usually applied *sparsely* (Section 2.3) and *locally* (Section 2.4).

## 2.2 ARCHITECTURE OF SPARSE AND LOCAL NEURAL LOGIC MACHINES

Fig. 1 shows the overall architecture of SpaLoc, which can be seen as a version of NLMs based on sparse tensors. A SpaLoc model is composed of multiple *relational reasoning layers* (RRLs) that operate on hyperedge representations. It also shows the detailed computation graph at a single layer in a NLM with 3-ary relations. The EXPAND operation propagates representations from lower-arity tensors to a higher-arity form (e.g., from each node to the edges connected to it). The REDUCE operation aggregates higher-arity representations into a lower-arity form (e.g., aggregate the information from all edges connected to a node into that node). The PERMUTE operation fuses the representations of hyperedges that share the same set of entities but in different order, such as $(A, B)$ and $(B, A)$. Finally, NN is a linear layer with nonlinear activation that computes the representation for the next layer. The neural network module NN is applied independently and identically to all tuples of the same arity. This ensures that a NLM learned on a small universe can be applied to a larger universe of entities.

Our RRLs are based on sparse tensor representations. In the following, we let $N = |\mathcal{V}|$ be the number of entities (nodes) in the hypergraph, $\mathcal{V} = \{o_1, o_2, \cdots o_N\}$ be the entities, $x_1, x_2, \cdots$ be variables, and $f(x_1, x_2, \cdots, x_r)$ be a mapping from a tuple $(x_1, x_2, \cdots, x_r)$ to a vector representation associated with that tuple. We will denote the maximum arity of tensors considered in a SpaLoc model as $R$.

**Sparse tensor representation of hyperedge representations.** In SpaLoc, the vector representations $f(x_1, x_2, \cdots, x_r)$ associated with all hyperedges of arity $r$ are represented as a coordinate-list (COO) format sparse tensor. That is, each tensor is represented as two tensors $\mathcal{F} = (\mathbf{I}, \mathbf{V})$, each with $M$ entries. The first tensor $\mathbf{I}$ is an *index* tensor, of shape $M \times r$, in which each row denotes a tuple $(x_1, x_2, \cdots, x_r)$. The second tensor $\mathbf{V}$ is a *value* tensor, of shape $M \times D$, where $D$ is the length of $f(x_1, x_2, \cdots, x_r)$. Each row $\mathbf{V}[i]$ denotes the vector representation associated with the tuple $\mathbf{I}[i]$. For all tuples that are not recorded in $\mathbf{I}$, their representations are treated as all-zero vectors. As an example, in Fig. 2, the input relationship *parent*$(x, y)$ is represented as a sparse tensor with three entries.

**Sparse relational reasoning layers.** Now, we show how each of the critical operations of an NLM can be performed on this sparse representation.

The EXPAND operation takes a sparse tensor $\mathcal{F}$ of arity $r$ and creates a new sparse tensor $\mathcal{F}'$ with arity $r + 1$: it propagates the representation from arity $r$ to arity $(r + 1)$, such as from nodes to edges. This is implemented by duplicating each entry $f(x_1, \cdots, x_r)$ in $\mathcal{F}$ by $N$ times, creating the $N$ new vector representations for $(x_1, \cdots, x_r, o_i)$ for all $i \in \{1, 2, \cdots, N\}$. Fig. 2 gives an example.

The REDUCE operation takes a sparse tensor $\mathcal{F} = (\mathbf{I}, \mathbf{V})$ of arity $r$ and creates a new sparse tensor $\mathcal{F}'$ with arity $r - 1$: it aggregates all information associated with all $r$-tuples: $(x_1, x_2, \cdots, x_{r-1}, ?)$ with the same $r - 1$ prefix. In SpaLoc, the aggregation function is chosen to be *max*. Thus,

$$f'(x_1, \cdots, x_{r-1}) = \max_{z: (x_1, \cdots, x_{r-1}, z) \in \mathbf{I}} f(x_1, \cdots, x_{r-1}, z).$$

Fig. 2 gives an example.

The PERMUTE operation takes a sparse tensor $\mathcal{F}$ of arity $r$ and creates a new sparse tensor $\mathcal{F}'$ of the same arity. However, the length of the vector representation will grow from $D$ to $D' = r! \times D$. It fuses the representation of hyperedges that share the same set of entities. Mathematically,

$$f'(x_1, \cdots, x_r) = \operatorname*{Concat}_{(x'_1, \cdots, x'_r) \text{ is a permutation of } (x_1, \cdots, x_r)} [f(x'_1, \cdots, x'_r)].$$

If a permutation of $(x_1, \cdots, x_r)$ does not exist in $\mathcal{F}$, it will be treated as an all-zero vector. Thus, the number of entries $M$ may increase or remain unchanged. Fig. 2 gives an example.

The $i$-th sparse relational reasoning layer has $R + 1$ linear layers $L^{(i,0)}, L^{(i,1)}, \cdots, L^{(i,R)}$ with nonlinear activations (e.g., ReLU) as submodules with arities 0 through $R$. For each arity $r$, we will concatenate the feature tensors expanded from arity $r - 1$, reduced from arity $r + 1$, and the output from the previous layer, apply a permutation, and apply $L^{(i,r)}$ on the derived tensor. In the concatenation step, since tensors from different sources may have different sets of entries, we will match the entries for different tensors with the necessary all-zero tensors.

We have seen how a sparse representation for NLMs can support efficient inference. In the next sections, we show how to learn sparse NLMs from graphs, and then how to use subsampling to learn efficiently from very large training graphs.

## 2.3 Sparsification through Hoyer Regularization

SpaLoc achieves sparsity during training by adding regularization terms to entries that are not useful in making the target prediction. At each layer, for each hyperedge $(x_1, \cdots, x_r)$, our model jointly predicts a vector representation $f(x_1, \cdots, x_r)$ as well as a hyperedge gate $g(x_1, \cdots, x_r) \in [0, 1]$ During training, we modulate the input representation $f$ with the gate value $g$; during inference, we prune all hyperedges whose gate value is smaller than a scalar hyperparameter $\epsilon = 5\%$. We augment the task loss with a sparsity regularization loss that encourages the network to prune as many edges as possible.

**Gated hyperedge representation.** Recall that each sparse relational reasoning layer is composed of $R + 1$ linear layers. In SpaLoc, for each linear layer $L^{(i,r)}$, we add linear gating layer, $L_g^{(i,r)}$, which has sigmoid activation and outputs a scalar value in range $[0, 1]$ that can be interpreted as the importance score for each hyperedge. During training, we modulate the output of $L^{(i,r)}$ with this importance value. Specifically, the output of layer $i$ arity $r$ is $L^{(i,r)}(\mathcal{F}) \odot L_g^{(i,r)}(\mathcal{F})$, where $\mathcal{F}$ is the

input sparse tensor, and $\odot$ is the element-wise multiplication operation. During inference, we prune out edges with a small importance score.

**Hoyer-Square measure.** We use a metric inspired by the Hoyer measure (2004) for quantifying the sparsity of the hyperedges and as the regularization loss to encourage hyperedge sparsity. It is based on the $L_1$ norm and the $L_2$ norm of a vector. Let $x$ be an arbitrary vector of length $n$. Then

$$Hoyer(x) = \frac{(\sum_i^n |x_i|)/\sqrt{\sum_i^n x_i^2} - 1}{\sqrt{n} - 1}$$

The Hoyer measure takes values from 0 to 1. The larger the Hoyer measure of a tensor, the denser the tensor is. In order to assign weights to different tensors based on their size, we use the Hoyer-Square measure (Yang et al., 2020),

$$H_S(x) = \frac{(\sum_i^n |x_i|)^2}{\sum_i^n x_i^2},$$

which ranges from 1 (sparsest) to $n$ (densest). Intuitively, the Hoyer-Square measure is more suitable than $L_1$ or $L_2$ regularizers for graph sparsification since it encourages large values to be close to 1 and others to be zero, i.e., extremity. It has been widely used in sparse neural network training and empirically proved to have better performance than other sparse measures Hurley & Rickard (2009). We also empirically compare $H_S$ with other sparsity measures in Appendix C.

The overall training objective of SpaLoc is the task objective plus the sparsification loss, $\mathcal{L} = \mathcal{L}_{task} + \lambda\mathcal{L}_{density}$, where $\mathcal{L}_{density}$ is the sum of the $H_S$ for all gate tensors across all layers, divided by the sum of the size of these tensors:

$$\mathcal{L}_{density} = \frac{\sum_i \sum_{r=0}^R H_S(g^{(i,r)})}{\sum_i \sum_{r=0}^R \#g^{(i,r)}}, \; g^{(i,r)} = L_g^{(i,r)}(\mathcal{F})$$

where $i$ sums over all layers of the SpaLoc and $r$ sums over all arities and # counts the number of elements in a tensor.

## 2.4 SUBGRAPH TRAINING

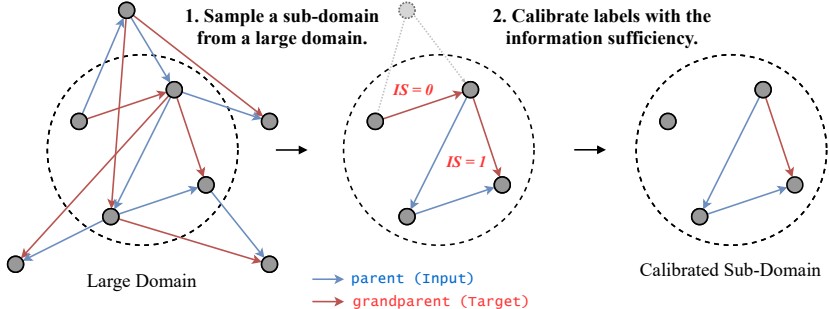

Figure 3: The subgraph training contains two steps. First, we sample a subset of nodes from a large input graph. Next, we calibrate the labels for edges in the sub-sampled graph. Note that the edge with *information sufficiency* 0 will be labeled as a negative example in this subgraph.

Regularization enables us to learn a sparse model that will be efficient at inference, but does not address the problem of *learning* efficiently from large training graphs. In this section we describe a novel strategy for substantially reducing training complexity, based on the observation that an inferred relation among a set of entities generally depends only on a small set of other entities that are "between" the target entities in the hypergraph, in the sense that they are connected via short paths of relevant relations. To exploit this observation, we employ a sub-graph sampling procedure during both training and inference. In the following, we first present a information-theoretic measure to quantify the sufficiency of information in a sub-sampled graph for determining the relationship between two entities, namely, *information sufficiency*. Next, we present a sub-graph sampling procedure that is designed to maximize the information sufficiency for training and inference. Finally, since it is inevitable that we will lose some information during the sampling procedure, we further propose a training paradigm that calibrates the labels for training examples based on the information sufficiency.

**Information sufficiency.** Let $\mathcal{H}_S = (\mathcal{V}_S, \mathcal{E}_S)$ be a sub-hypergraph of hypergraph $\mathcal{H} = (\mathcal{V}, \mathcal{E})$, and $e^* = (y_1, \ldots, y_r)$ be a target hyperedge in $\mathcal{H}_S$, where $y_1, \cdots, y_r \in \mathcal{V}_S \subset \mathcal{V}$. Intuitively, in order to determine the label for this hyperedge, we need to consider all "paths" that connect the nodes $\{y_1, \ldots, y_r\}$. More formally, we say a sequence of $K$ hyperedges

$$\underbrace{(x_1^1, \cdots, x_{r_1}^1)}_{e_1}, \underbrace{(x_1^2, \cdots, x_{r_2}^2)}_{e_2}, \cdots, \underbrace{(x_1^K, \cdots, x_{r_k}^K)}_{e_K},$$

is a *hyperedge path* for nodes $\{y_1, \cdots y_r\}$ if and only if $\{y_1, \cdots y_r\} \subset \bigcup_{j=1}^{K} e_j$ and $e_j \cap e_{j+1} \neq \emptyset$ for all $j$. In a graph with only binary edges, this is equivalent to the existence of a path from one node $y_1$ to another node $y_2$. We define the *information sufficiency* measure for a hyperedge $e^*$ in subgraph $\mathcal{H}_S$ as the fraction of the total number of paths connecting $\{y_i, \cdots, y_r\}$ that are retained in the subgraph:

$$IS\left((y_1, \cdots, y_r) \mid \mathcal{H}_S, \mathcal{H}\right) := \frac{\#\text{Paths connecting } (y_1, \cdots, y_r) \text{ in } \mathcal{H}_S}{\#\text{Paths connecting } (y_1, \cdots, y_r) \text{ in } \mathcal{H}} {}_* \quad (1)$$

In practice, we approximate *IS* by only counting the number of paths whose length is less than a task-dependent threshold $\tau$ for efficiency. The number of paths in a large graph can be pre-computed and cached before training, and the overhead of counting paths in sampled graph is small, so this computation does not add much overhead to training and inference. In the case that the input graphs have maximum arity 2, paths can be counted efficiently by taking powers of the graph adjacency matrix.

**Subgraph sampling.** We use different sub-graph sampling strategies for training and inference.

During training, each data point is a tuple $(\mathcal{H}, f, f')$ where $\mathcal{H}$ is the input graph, $f$ is the input representation, and $f'$ is the desired output representation. We will sample a subgraph $\mathcal{H}' \subset \mathcal{H}$, and train models to predict the value of $f'$ on $\mathcal{H}'$ given $f$. For example, we will train models to predict the *grandparent* relationship between all pairs of entities in $\mathcal{H}'$ based on the *parent* relationship between entities in $\mathcal{H}'$. Thus, our goal is to find a subgraph that retains most of the paths connecting nodes in this subgraph. We achieve this using a *neighbor expansion sampler* that uniformly samples a few nodes from $\mathcal{V}$ as the seed nodes. It then samples new nodes connected with one of the nodes in the graph into the sampled graph and runs this "expansion" procedure for multiple iterations to get $\mathcal{V}_S$. Finally, we include all edges that connect nodes in $\mathcal{V}_S$ to form the final subsampled hypergraph.

During inference, usually our goal is to just infer the relationship between one pair of entities $f'(y_1, y_2)$ on a graph $\mathcal{H}$ given the input representation $f$. In this case, we will use a *path sampler* when dealing with large domains, which samples paths connecting $y_1$ and $y_2$ and induce a subgraph from these paths. We provide ablation studies comparing different sampling techniques in Section 3.4. The implementation details of our information sufficiency computation and subgraph samplers can be found in Appendix D and Appendix B.

**Training label calibration with IS.** Due to the information loss caused by graph subsampling, the closed-world assumption (CWA) no longer holds in the subgraph. That is, the information contained in the subgraph may not be sufficient to make predictions about a target relationship. For example, in a family relationship graph, removing a subset of nodes may cause the system unable to conclude whether a specific person $x$ has a sibling.

Thus, we propose to calibrate the model training by assigning each example $f'(y_1, \cdots, y_r)$ with a soft label. Specifically, consider a binary classification task $f'$. That is, function $f'$ is a mapping from a hyperedge tuple of arity $r$ to $\{0, 1\}$. Denote the model prediction as $\hat{f}'$. Typically, we will train the SpaLoc model with a binary cross-entropy loss between $\hat{f}'$ and the groundtruth $f'$. In our subgraph training, we will instead compute a binary cross-entropy loss between $\hat{f}'$ and $f'_{\mathcal{H}_S} \odot IS$, where $\mathcal{H}_S$ is the sub-sampled graph. Mathematically,

$$\left(f'_{\mathcal{H}_S} \odot IS\right)(y_1, \cdots, y_r) \triangleq f'_{\mathcal{H}_S}(y_1, \cdots, y_r) \cdot IS\left((y_1, \cdots, y_r) \mid \mathcal{H}_S, \mathcal{H}\right).$$

Fig. 3 shows a concrete example of the label calibration process.

---

${}^*\frac{0}{0}$ is defined as 1.

Table 1: Results (Per-class Accuracy) on family tree reasoning benchmarks. Models are tested on domains with 100 objects. Minus mark means the model cannot scale up in training datasets of corresponding sizes or cannot handle ternary predicates.

| Family Tree | MemNN | | $\partial$ILP | | NLM | | GraIL (R-GCN) | | SpaLoc (Ours) | |
|---|---|---|---|---|---|---|---|---|---|---|
| $N_{\text{train}}$ | 20 | 2,000 | 20 | 2,000 | 20 | 2,000 | 20 | 2,000 | 20 | 2,000 |
| HasFather | 65.24 | - | 100 | - | 100 | - | 100 | 100 | 100 | 100 |
| HasSister | 66.21 | - | 100 | - | 100 | - | 97.05 | 97.95 | 100 | 97.97 |
| Grandparent | 64.57 | - | 100 | - | 100 | - | 99.95 | 98.08 | 100 | 100 |
| Uncle | 64.82 | - | 100 | - | 100 | - | 97.87 | 96.50 | 100 | 100 |
| MGUncle | 80.93 | - | 100 | - | 100 | - | 54.67 | 71.29 | 100 | 100 |
| Family-of-Three | - | - | - | - | 100 | - | - | - | 100 | 100 |
| Three-generations | - | - | - | - | 100 | - | - | - | 100 | 100 |

## 3 EXPERIMENTS

In this section, we compare SpaLoc with other methods in two aspects: accuracy and efficiency on large domains. We compare SpaLoc with other approaches on relational reasoning and real-world knowledge graph reasoning benchmarks. All experiments are under the inductive, supervised learning setup, i.e., testing objects and relations have not been seen during training, and the sizes of domains used for training and evaluation are also very different.

### 3.1 BASELINES

For the family tree reasoning tasks, we compare SpaLoc against four baselines. The first three are Memory Networks (MemNN) (Sukhbaatar et al., 2015), $\partial$ILP (Evans & Grefenstette, 2018) and Neural Logic Machines (NLMs; Dong et al., 2019), which are state-of-the-art models for relational rule learning tasks. For these models, we follow the configuration and setup in Dong et al. (2019). The fourth baseline is an inductive link prediction method based on graph neural networks, GraIL (Teru et al., 2020). Since GraIL can be only used for link prediction, we use the full-batch R-GCN (Schlichtkrull et al., 2018), the backbone network of GraIL, for node property predictions. For inductive knowledge graph reasoning datasets, we compare SpaLoc with state-of-the-art models, including Neural LP (Yang et al.), DRUM (Sadeghian et al.), RuleN (Meilicke et al., 2018), GraIL and TACT (Chen et al., 2021). For transductive learning tasks, we compare SpaLoc with four representative knowledge graph embedding methods, including TransE (Bordes et al., 2013), DistMult (Yang et al., 2015), ComplEx (Trouillon et al., 2017), and RotatE (Sun et al., 2019).

### 3.2 FAMILY TREE REASONING

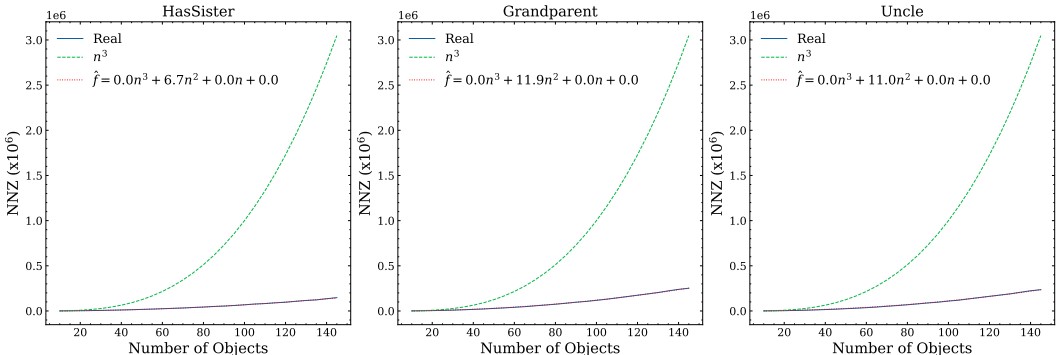

Figure 4: Numbers of non-zero elements (NNZ) in the intermediate groundings of SpaLoc *vs*. the number of objects in the evaluation domains.

We evaluate SpaLoc on the family-tree reasoning benchmark, a synthetic benchmark for inductive logic programming. The goal of family-tree tasks is to induce target family relationships or member properties given the following binary relations: Son, Daughter, Father, and Mother. We give concrete logical definitions of the family tree benchmark in the Appendix E.

**Accuracy & Efficiency**. From Table 1 we observe the superiority of SpaLoc in terms of accuracy and efficiency. SpaLoc achieves perfect accuracy on test set. And when handling large input domains

Table 2: Results (AUC-PR) on real-world knowledge graph inductive reasoning datasets from GraIL.

| Model | WN18RR | | | | FB15k-237 | | | | NELL-995 | | | |
|---|---|---|---|---|---|---|---|---|---|---|---|---|
| | v1 | v2 | v3 | v4 | v1 | v2 | v3 | v4 | v1 | v2 | v3 | v4 |
| Neural-LP | 86.02 | 83.78 | 62.90 | 82.06 | 69.64 | 76.55 | 73.95 | 75.74 | 64.66 | 83.61 | 87.58 | 85.69 |
| DRUM | 86.02 | 84.05 | 63.20 | 82.06 | 69.71 | 76.44 | 74.03 | 76.20 | 59.86 | 83.99 | 87.71 | 85.94 |
| RuleN | 90.26 | 89.01 | 76.46 | 85.75 | 75.24 | 88.70 | 91.24 | 91.79 | 84.99 | 88.40 | 87.20 | 80.52 |
| GraIL | 94.32 | 94.18 | 85.80 | 92.72 | 84.69 | 90.57 | 91.68 | 94.46 | 86.05 | 92.62 | 93.34 | 87.50 |
| TACT-base | 98.11 | 97.11 | 88.34 | 97.25 | 87.36 | 94.31 | 97.42 | 98.09 | 94.00 | 94.44 | 93.98 | 94.93 |
| TACT | 96.15 | 97.95 | 90.58 | 96.15 | 88.73 | 94.20 | 97.10 | 98.30 | 94.87 | 96.58 | 95.70 | 96.12 |
| SpaLoc | **98.18** | **99.83** | **96.66** | **99.30** | **99.73** | **99.38** | **99.53** | **99.39** | **100** | **98.27** | **96.19** | **97.37** |

($N_{\text{train}} = 2000$), SpaLoc works well because of the sampling techniques, while most baselines cause out-of-memory errors.

**Sparsity**. In Fig. 4, we show the number of non-zero elements in SpaLoc's intermediate groundings versus the size of input domains on `HasSister`, `Grandparent` and `Uncle`. We fit a cubic polynomial equation $\hat{f}$ to the data points to illustrate the learned inference complexity of SpaLoc. The SpaLoc we use in these three datasets has an arity of 3 so the maximum capacity of the model is $\Theta(N^3)$. However, the memory complexity of SpaLoc converges to the optimal algorithm complexity of these tasks , which is $O(N^2)$. In comparison, the memory complexity of original dense NLMs will be fixed to $\Theta(N^3)$ when the model architecture is fixed.

### 3.3 REAL-WORLD KNOWLEDGE GRAPH REASONING

We evaluate SpaLoc on the real-world knowledge-graph inductive reasoning benchmarks proposed in GraIL, whose training and evaluation sets are disjoint sub-graphs extracted from WN18RR (Dettmers et al., 2018), FB15k-237 (Toutanova et al., 2015), and NELL-995 (Xiong et al., 2017). For each knowledge graph, there are four versions of inductive datasets with increasing sizes.

Table 3: Results (AUC-PR) of transductive link prediction on real-world knowledge graphs.

| | WN18RR | NELL-995 | FB15K-237 |
|---|---|---|---|
| TransE | 93.73 | 98.73 | 98.54 |
| DistMult | 93.08 | 97.73 | 97.63 |
| ComplEx | 92.45 | 97.66 | 97.99 |
| RotatE | 93.55 | 98.54 | 98.53 |
| GraIL | 90.91 | 97.79 | 92.06 |
| **SpaLoc** | **96.76** | **99.27** | **99.61** |

To further demonstrate the scalability of SpaLoc, we also apply SpaLoc to the complete real knowledge graphs. SpaLoc still uses the inductive learning setting. That is, it does not use any entity information learned from training, only using the structural information for inference. We use the standard WN18RR, FB15K-237, and NELL-995 benchmarks. We follow the original data split.

Table 2 and Table 3 show the results. SpaLoc significantly outperforms all baselines on all datasets and achieves a new state-of-the-art. This demonstrates SpaLoc's excellent scalability and generalization ability on noisy large-scale real-world data.

### 3.4 ABLATION STUDY

Table 4: Comparison of different samplers.

Table 5: Comparison (per-class accuracy) for different label calibration methods.

| $N_s/N$ | Node | | Walk | | Neighbor | |
|---|---|---|---|---|---|---|
| | Acc | MIS | Acc | MIS | Acc | MIS |
| 20 / 50 | 100 | 54.82 | 100 | 85.14 | 100 | 89.78 |
| 20 / 200 | 100 | 33.05 | 100 | 71.51 | 100 | 80.60 |
| 20 / 500 | 58.18 | 27.27 | 100 | 78.22 | 100 | 78.70 |
| 20 / 1,000 | 1.84 | 24.49 | 100 | 77.18 | 100 | 78.38 |
| 20 / 2,000 | 0 | 19.66 | 100 | 79.69 | 100 | 78.53 |

| Sampler | HasFather | | | HasSister | | |
|---|---|---|---|---|---|---|
| | - | LS | IS | - | LS | IS |
| Node | 50.00 | 50.00 | 50.00 | 50.00 | 50.00 | **80.72** |
| Walk | 50.00 | 52.41 | **100** | 59.90 | 75.13 | **93.16** |
| Neighbor | 50.00 | 51.63 | **100** | 75.29 | 78.06 | **97.97** |

**Subgraph sampling.** Beside the *neighbor expansion sampler*, there are also some intuitive and efficient sub-graph samplers, such as the random node (`Node`) and random walk (`Walk`) samplers proposed by Zeng et al. (2020). We compare these samplers in two ways. Empirically, we test the models trained with the samplers. Theoretically, we evaluate the overall information sufficiency of a sub-graph by averaging the information sufficiency of all target edges, introducing a metric called Mean Information Sufficiency (MIS).

In Table 4, we show the results on the `Grandparent` dataset. The `Node` sampler does not leverage locality, so the performance of models and MIS drops with the size of whole graph increases. SpaLocs

trained with `Walk` and `Neighbor` samplers achieve perfect accuracies, and the MIS of sub-graphs provided by these sampers are also comparable. Also, the MIS of `Walk`'s and `Neighbor`'s sub-graphs do not drop significantly with the increasing size of the whole graph, which ensures the application of these samplers on arbitrarily large graphs.

**Label calibration.** We compare our information sufficiency based label calibration method against a simple constant label smoothing. We show our results in Table 5. In the table, "LS" means constant label smoothing by multiplying positive labels with a constant $\alpha = 0.9$, and "IS" means label calibration with the information sufficiency. The impact of sampling and the failure of the closed-world assumption is particularly significant on the two datasets `HasFather` and `HasSister`. Without label calibration, the performance of the model trained on the sub-graphs is close to random prediction. Our information-based calibration method improves the performance by a large margin, and is significantly superior to naive smoothing techniques.

## 4 RELATED WORK

**(Hyper-)Graph representation learning**. (Hyper-)Graph representation learning methods, including message passing neural networks (Shervashidze et al., 2011; Kipf & Welling, 2017; Velickovic et al., 2018; Hamilton et al., 2017) and embedding based methods (Dettmers et al., 2018; Toutanova et al., 2015), have been widely used for knowledge discovery. Since these methods treat relations (edges) as fixed indices for node feature propagation, their computational complexity is usually small (e.g., $O(NE)$), and they can be applied to large datasets. However, the fixed relation representation and low complexity restrict the expressive power of these methods (Xu et al., 2019; 2020) so they cannot be used to solve some general complex problems, such as inducing tenary relational rules. Moreover, some of the widely used methods such as knowledge embeddings are inherently transductive, and cannot learn lifted rules that generalize to unseen domains. By contrast, relations in SpaLoc are evolving for reasoning, and the architecture of SpaLoc can be adjusted for problems of different complexities. Therefore, SpaLoc can learn lifted and complex rules.

**Inductive rule learning**. In addition to graph learning frameworks, many previous approaches have studied how to learn generalized rules from data (Muggleton, 1991; Friedman et al., 1999), i.e., inductive logic programming (ILP), with recent work integrating neural networks into ILP systems to combat noisy and ambiguous inputs (Dong et al., 2019; Evans & Grefenstette, 2018; Sukhbaatar et al., 2015). However, due to the large search space of target rules, computational and memory complexities of these models are too high to scale up to many large real-world domains. SpaLoc addresses this scalability problem by leveraging the sparsity and locality of real-world rules, and thus can induce knowledge with local computations and optimal complexities.

**Efficient training and inference methods**. There is a rich literature on efficient training and inference of neural networks. Two directions that are relevant to us are model sparsification and sampling training. Han et al. (2016) propose to prune and compress the weights of neural networks for efficiency, and Yang et al. (2020) adopt Hoyer-Square regularization to sparsify models. SpaLoc extend this sparsification idea by adding regularization at intermediate sparse tensor groundings to encourage concise induction. Chiang et al. (2019) and Zeng et al. (2020) propose to sample sub-graphs for GNN training, and Teru et al. (2020) propose to construct sub-graphs for link prediction. SpaLoc generalizes these sampling methods to hypergraphs, and proposes the information sufficiency based calibration method to remedy the information loss introduced by sub-sampling.

## 5 CONCLUSION

We present SpaLoc, a framework for efficient training and inference of hypergraph reasoning networks. SpaLoc uses sparse tensors to represent high-order relationships (hyperedges), and implements finite-domain quantification to infer new knowledge. SpaLoc leverages sparsity and locality to train and infer efficiently. Through regularizing intermediate representation by a sparsification loss, SpaLoc's inference complexities on family tree tasks are the same as those of algorithms designed by human experts. To be applied to large datasets, SpaLoc sample sub-graphs for training and inference, and calibrate training labels with information sufficiency measure to alleviate information loss. SpaLoc generalizes well on large-scale relational reasoning benchmarks.

## REPRODUCIBILITY STATEMENT

To maximize reproducibility, we described our methodology in detail in Section 2 and our experimental setup in Section 3. All the data and codes in this paper will be released upon publication to facilitate future research.

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

# SUPPLEMENTARY MATERIAL

## A  EXPERIMENTAL CONFIGURATION

Table 6: Hyper-parameters for SpaLoc. The definition of depth and breadth are illustrated in Figure 1.

|  | Tasks | Depth | Breadth | Hidden Dims | Batch size | Subgraph size | $\tau$ |
|---|---|---|---|---|---|---|---|
| Family Tree | HasFather | 5 | 3 | 8 | 8 | 20 | 1 |
|  | HasSister | 5 | 3 | 8 | 8 | 20 | 2 |
|  | Grandparent | 5 | 3 | 8 | 8 | 20 | 2 |
|  | Uncle | 5 | 3 | 8 | 8 | 20 | 2 |
|  | MGUncle | 5 | 3 | 8 | 8 | 20 | 2 |
|  | Family-of-three | 5 | 3 | 8 | 8 | 20 | 2 |
|  | Three-generations | 5 | 3 | 8 | 8 | 20 | 2 |
| Inductive KG | WN18RR | 6 | 3 | 64 | 128 | 10 | 3 |
|  | FB15K-237 | 6 | 3 | 64 | 64 | 20 | 3 |
|  | NELL-995 | 6 | 3 | 64 | 128 | 10 | 3 |
| Transductive KG | WN18RR | 6 | 3 | 64 | 64 | 20 | 3 |
|  | FB15K-237 | 6 | 3 | 64 | 64 | 20 | 3 |
|  | NELL-995 | 6 | 3 | 64 | 64 | 20 | 3 |

We optimize all models with Adam (Kingma & Ba, 2015) and use an initial learning rate of 0.005. All experiments are under the supervised learning setup, we use Softmax-Cross-Entropy as loss function.

Table A shows hyper-parameters used by SpaLoc. For all MLP inside SpaLoc, we use no hidden layer and the sigmoid activation. Across all experiments in this paper, the maximum arity of intermediate predicates (i.e., the "breadth") is set to 3 as a hyperparameter, which allows SpaLoc to realize all first-order logic (FOL) formulas with at most three variables, such as a "transitive relation rule." We set the specification threshold $\epsilon$ to 0.05 in all our experiments. This value is chosen based on the validation accuracy of our model. In practice, we observe that any values between 0.01 and 0.1 do not significantly impact the performance of our method and the inference complexity. We also set the multiplier of the sparsification loss $\lambda$ to 0.01 in all SpaLoc's experiments.

## B  IMPLEMENTATION OF SUBGRAPH SAMPLERS

Both the neighbor expansion and the path sampler sample subgraphs by inducing from selected node sets. To deal with input hypergraphs with any arities, the samplers simplify the input hypergraphs into binary graphs. We define two nodes in the hypergraph are connected if they are covered by a hyperedge. Therefore, the neighbor expansion and path finding algorithms used by the samplers can be applied to any hypergraphs for finding node sets. After enough nodes are collected, the sampler will induce a sub-hypergraph from the original hypergraph by preserving all of the hyperedges lying in the set.

## C  ABLATION STUDY ON SPARSIFICATION LOSS

Table 7: Comparison of different sparsification losses.

|  | HasSister | | Grandparent | | Uncle | |
|---|---|---|---|---|---|---|
|  | Accuracy | Density | Accuracy | Density | Accurcay | Density |
| $L_1$ | 91.81 | 0.48% | 99.8% | 0.99% | 74.69% | 0.68% |
| $L_2$ | 100 | 0.75% | 100% | 0.61% | 94.46% | 2.44% |
| $H_S$ | 100 | 0.51% | 100% | 0.48% | 100% | 0.87% |

We compare our Hoyer-Square sparsification loss against the $L_1$ and $L_2$ regularizers on the family tree datasets. In Table 7, we show the performance of SpaLoc trained with different sparsification losses. All models are tested on domains with 100 objects.

"Density" is the percentage of non-zero elements (NNZ) in the intermediate groundings. The lower the density, the better the sparsification and the lower the memory complexity. We can see that, compared with $L_1$ and $L_2$ regularizers, the Hoyer-Square loss yields a higher or comparable sparsity while maintaining the nearly perfect accuracy.

## D    CALCULATION OF THE INFORMATION SUFFICIENCY

The crucial part in the computation of the information sufficiency is to count the number of $k$-hop hyperpaths connecting a given set of nodes $\{v_1, \ldots, v_r\}$ in a hypergraph. We use the incidence matrix to calculate this. Firstly, we use a $n \times m$ incidence matrix $B$ to represent the hypergraph $\mathcal{H} = (\mathcal{V}, \mathcal{E})$, where $n = |\mathcal{V}|$ and $m = |\mathcal{E}|$, such that $B_{ij} = 1$ if the vertex $v_i$ and edge $e_j$ are incident and 0 otherwise. Next, $B^{(k)} := (BB^T)^{k-1}B$ is the $k$-hop incidence matrix of the hypergraph, i.e., $B_{ij}^{(k)}$ is the number of $k$-hop paths that the vertex $v_i$ and edge $e_j$ are incident.

For example, when $r = 2$, there are $B_i^{(k)}B_j^T$ $k$-hop paths connecting vertex $v_i$ and $v_j$. When $r = 1$, there are $\sum_j B_{ij}^{(k)}$ $k$-hop paths connecting to vertex $v_i$.

## E    DEFINITION OF RELATIONS IN FAMILY TREE

The inputs predicates are: `Father`$(x, y)$, `Mother`$(x, y)$, `Son`$(x, y)$, `Daughter`$(x, y)$.

The target predicates are:

- `HasFather`$(x) := \exists a \, $`Father`$(x, a)$
- `HasSister`$(x) := \exists a \exists b \, $`Father`$(x, a) \wedge $`Daughter`$(a, b) \wedge \neg(b = x)$
- `Grandparent`$(x, y) := \exists a \, $`parent`$(x, a) \wedge $`parent`$(a, y)$
  `parent`$(x, y) := $`Father`$(x, y) \vee $`Mother`$(x, y)$
- `Uncle`$(x, y) := \exists a \, $`Grandparent`$(x, a) \wedge $`Son`$(a, y) \wedge \neg$`Father`$(x, y)$
- `MGUncle`$(x, y) := \exists a \exists b \, $`Grandmother`$(x, a) \wedge $`Mother`$(a, b) \wedge $`Son`$(b, y)$
  `Grandmother`$(x, y) = \exists a \, $`Parent`$(x, a) \wedge $`Mother`$(a, y)$
- `Family-of-three`$(x, y, z) = $`Father`$(x, y) \wedge $`Mother`$(x, z)$
- `Three-generations`$(x, y, z) = $`Parent`$(x, y) \wedge $`Parent`$(y, z)$

We follow the dataset generation algorithm presented in Dong et al. (2019). In detail, we simulate the growth of families to generate examples. For each new member of the family, we randomly assign a gender and a pair of parents (can be none, which means it the oldest person in the family tree) for it. After the family tree is generated, we label the relationships according to the definitions above.

## F    EFFICIENCY ANALYSIS

We compare the time and memory complexity of SpaLoc against NLM on the `Has-Sister` and `Grandparent` tasks. In Fig. 5, we plot the curve of memory consumption and inference time of each sample as a function of the number of objects in the evaluation domains. The experimental results show that our method can reduce the space complexity from the original $O(n^3)$ complexity of NLM to the $O(n^2)$ complexity required to solve the has-sister and grandparent task. In terms of the inference time, there is a constant level of optimization. On small graphs, adding sparsity does not necessarily improve the running time on GPUs because of the overhead of handling sparse tensors due to the implementation. More efficient implementation of sparse operations is interesting future work.

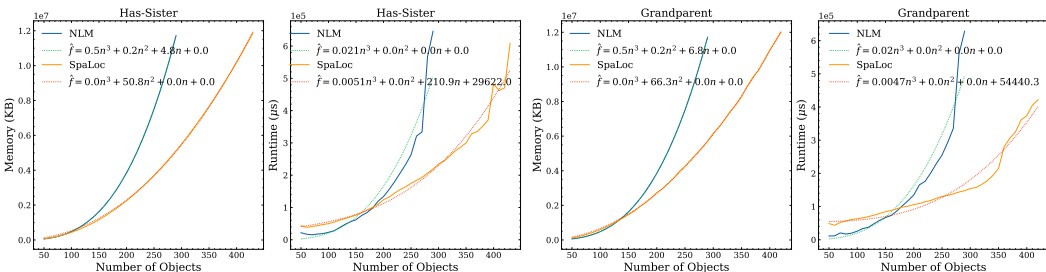

Figure 5: The memory consumption and the inference time of each sample *vs.* the number of objects in the evaluation domains.

## G  IMPLEMENTATION OF SPALOC IN PYTORCH

The following python code is a minimal implementation for a single layer SpaLoc with PyTorch and
[†]`torch_scatter`. The syntax is highlighted and is best viewed in color.

```python
from itertools import permutations
import SparseTensor
import torch
from torch import nn
import math
from torch_sparse import scatter

def expand(input: SparseTensor):
    """Expands a new dimension in input by duplicating each entry
    ↪  for n times"""
    coo = (input.coo.unsqueeze(-1) * input.n) +
    ↪  torch.arange(input.n).unsqueeze(0)
    val = input.val.unsqueeze(1).expand(input.length, input.n,
    ↪  input.channel)
    val = val.contiguous().view(-1, input.channel)
    return SparseTensor(val, input.n, input.arity + 1,
    ↪  input.channel, coo=coo.flatten())

def reduce(input: SparseTensor):
    """Reduces max at the last dimension,
        except for elements whose coordinates contain duplicate
↪ objects.
    """
    val = input.val.clone()
    val[reduce_mask(input)] = -math.inf
    flatten_index = input.coo // input.n
    coo, inverse_index = torch.unique(flatten_index,
    ↪  return_inverse=True, sorted=True)
    val = scatter(val, inverse_index, dim=0, reduce='max')
    return SparseTensor(val, input.n, input.dim - 1,
    ↪  input.channel, coo=coo)

def neural_logic(input, hidden_dim):
    """An MLP layer applied on permutations of the input."""
    return MLP(permute(input), hidden_dim,
    ↪  activation=nn.Sigmoid())

def spaloc_breath3(input0, input1, input2, input3, hidden_dim):
    """A SpaLoc layer with breath 3.
    Args:
      input0: SparseTensor of shape [hidden_dim], nullary
↪ predicates.
      input1: SparseTensor of shape [N, hidden_dim], unary
↪ predicates.
      input2: SparseTensor of shape [N, N, hidden_dim], binary
↪ predicates.
```

---

[†][https://github.com/rusty1s/pytorch_scatter](https://github.com/rusty1s/pytorch_scatter)

```
40          input3: SparseTensor of shape [N, N, N, hidden_dim], tenary
    ↪   predicates.
41          hidden_dim: int, hidden dimension.
42      Returns:
43          4 SparseTensors, output nullary, unary, binary tenary
    ↪   predicates respectively.
44      """
45      agg0 = concat([input0, reduce(input1)])
46      agg1 = concat([input1, expand(input0), reduce(input2)])
47      agg2 = concat([input2, expand(input1), reduce(input3)])
48      agg3 = concat([input3, expand(input2)])
49      outputs = [neural_logic(x, hidden_dim) for x in [agg0, agg1,
        ↪   agg2, agg3]]
50      return outputs
51
52
53  def reduce_mask(input):
54      mask = torch.zeros(input.length, dtype=torch.bool)
55      coo = input.coo
56      for i in range(input.arity):
57          for j in range(i+1, input.dim):
58              mask = torch.logical_or(mask, coo[:, i] == coo[:, j])
59      return mask
60
61
62  def permute(input):
63      if input.dim <= 1:
64          return input
65      index = tuple(range(0, input.dim))
66      coo = input.coo()
67      weight = torch.tensor([[input.n ** i for i in
        ↪   reversed(range(input.dim))]])
68      ts = []
69      for p in permutations(index):
70          new_coo = (weight * coo[:, p]).sum(-1)
71          ts.append(SparseTensor(input.val, input.n,
72                                 input.dim, input.channel,
                                   ↪   coo=new_coo))
73      return input.cat(ts)
74
75  def concat(inputs):
76      nnzs = [t.length for t in inputs]
77      n = inputs[0].n
78      dim = inputs[0].dim
79      channel = sum([t.channel for t in inputs])
80      coo = torch.cat([t.coo for t in inputs], dim=-1)
81      coo, inverse_index = torch.unique(
82          coo, return_inverse=True, sorted=True)
83      val = torch.zeros((len(coo), channel))
84      current_channel, current_idx = 0, 0
85      for idx, t in enumerate(inputs):
86          next_channel, next_idx = current_channel + t.channel,
              ↪   current_idx + t.length
87          val[inverse_index[current_idx:next_idx],
88              current_channel:next_channel] = t.val
89          current_channel, current_idx = next_channel, next_idx
90      return SparseTensor(val, n, dim, channel, coo=coo)
```

