# OpenReview forum: "Efficient Training and Inference of Hypergraph Reasoning Networks"
_ICLR.cc/2022/Conference — ICLR 2022 Submitted_

### Official Review · Reviewer_QNYc · 2021-11-02

**Correctness:** 1
**Technical Novelty And Significance:** 3
**Empirical Novelty And Significance:** 2
**Recommendation:** 3
**Confidence:** 3

**Main Review:**

## Strengths and weaknesses
Strengths:
- The experiments in section 3.3 show strong predictive performance improvements over previous works
- Both the sparse tensor representation and the sub-sampling are interesting and sound methods for reducing the memory footprint of NLM during training
- The information sufficiency term for two nodes is computed as the fraction of hypergraph-paths that are retained in the sampled hypergraph, where a hypergraph-path generalizes the notion of paths in graphs to hypergraphs. This is a reasonable way to quantify the amount of information that is lost during sampling and the ablation demonstrates the effectiveness of using this during training.
- The paper is well written and legible for the most part.

Weaknesses:
- The premise of the paper is: training with many entities at once (20 vs 2000) is necessary or beneficial. This is not obvious, nor is it supported theoretically or empirically in the paper. For example, the experiment in section 3.2 compares against NLM by testing on 100 entities, when trained on 20 and 2000 entities. Since training with 20 entities already leads to perfect accuracy for NLM, it is not obvious why one would want to train with more. Furthermore, training the proposed model with more entities actually degrades the performance of the “HasSister” class.
- Section 3.3 “Real-world knowledge graph reasoning” does not describe the experimental setup, so I assume the experiments were done using the same setup as described in [Teru et al., 2020]. According to their description, this task is inductive in the entities only (nodes) and not the relations (edge types), meaning that there are no new relations in the targets. This setting is markedly different from the experiment in 3.2 and requires explicit mention and discussion.
Furthermore, it is not obvious why NLM would not work in section 3.3 by training on smaller input sizes and extrapolating to larger test sizes.
- Section 2.3 presents a method that encourages sparsity in the intermediate and output hypergraphs. The paper claims that this speeds up inference on large graphs, without any further explanation or validation. An ablation and further discussions on the trade-offs are necessary to motivate the importance of the methods presented in this section.
- The Hoyer-Square measure is presented as a better sparsification loss than the Hoyer, L1, and L2 alternatives, without any empirical support. An ablation would help convince the reader.
- The information sufficiency (IS) term takes the hyperparameter tau that determines the maximum path length that is still considered in the computation of IS. This appears to be a crucial value to set correctly and warrants further discussion on how to pick it and mentions of the concrete values used in the experiments.

## Additional Feedback and Questions
- Much of the paper is written with the assumption that the reader is familiar with the NLM paper, making it less accessible to a broader audience. Some example points that should be explained in the paper: What are lifted rules? What are the hyperedge feature domains considered in this work? What are the S and D typically? What is the arity of considered relations and the intermediate layers.
- There are many details on the method and the experimental setups missing that require explanation in the paper or appendix. For example: What is gF in “... in gF by N times, creating …”? How does M change with the PERMUTE operation? What are the f_hat’s in Figure 4?
- The proposed method appears to beat previous methods on most benchmarks, but it is not clear why. One way to provide more clarity is to describe the most significant differences of the proposed approach with the baselines.
- The introduction claims that the NLM scales exponentially with the number of entities and presents a cubic example O(N^3) (that is not exponential in N). The exponential scaling claim appears wrong and should be polynomial instead.
- Miscellaneous:
  - “Fig 1b”: there is no b
  - “L_g^(i,k)”: k should be r
  - “Let x be an arbitrary vector”: x_1, x_2, ... were already defined as the variables
  - “g^(i,r)”: this variable wasn’t introduced anywhere
  - “Sequence of k hyperedges”: mixing k and K here
  - Different notations (H’ and S) for the sampled hypergraph
  - “Label calibration..”: two dots


**Summary Of The Paper:**

This paper considers the task of inductive relation prediction: given a set of entities and relations, predict new types of relations. A specific example is: given a collection of people and the “is parent of” relation, predict the “is grandparent of” relation. The main contribution is a memory-efficient training and inference method for the Neural Logic Machines (NLM) [Dong et al., 2019]. For this, the paper presents a sparse tensor representation for the relations and add a loss term that encourages sparsity in intermediate layers and the output. Training on hypergraphs with many entities becomes computationally tractable (presumably on memory-limited GPUs), by sub-sampling the hypergraph (that represents the entities as nodes and relations as hyperedges). To deal with the situation in which vital nodes or hyperedges are removed during sampling, the paper introduces information sufficiency, a quantity used to scale the ground-truth labels.

**Summary Of The Review:**

I recommend rejecting the paper: It is currently not apparent why one would want to use many entities during training, when it’s possible to train with few entities and extrapolate to many at test time. Furthermore, the good results reported in section 3.3 seem irrelevant, since the experiment actually performs a different task from "relational rule induction".

The paper contributes to a specific line of research, by extending the NLM [Dong et al., 2019] to larger input sizes. In general, the paper would significantly benefit from more justifications and explanations for the proposed methods. More details on the experiments are also essential for the reader to understand the results. The technical contribution consists of a combination of different existing methods or variations thereof, and the information sufficiency value which appears to be new.

---

I thank the authors for the rebuttal and the additional ablation study and details on the hyperparameter tau. I keep my score at 3, because in my opinion the need of training with larger input graphs is insufficiently covered. In particular, the datasets reported in Table 2 are based on *sub-sampled* graphs, as described in Appendix F of [Teru et al., 2020]. This means that it is possible to train NLM on the KG datasets considered in this work, by simply applying the same sub-sampling strategy with fewer roots. The new results in Table 3 also do not support this point, since it looks like that training with the complete KG (instead of subgraphs v1, v2..) does not improve the performance.

---

> ### Author Response · Authors · 2021-11-23
> **Author Response to Reviewer QNYc**
>
> Thank you for acknowledging the performance and motivations of our work. We notice that our motivation and setup can be better clarified to address potential misunderstandings.  Please see our detailed answers below.
>
> ------
>
> Q1: The necessity of training on large domains.
>
> A1: The real data we want to work on, such as the real-world knowledge graphs considered in this paper (FB15K, NELL995), are large graphs (more than 10K nodes). Thus, developing techniques that can work with these large graphs is important.
> We leverage the advantage that NLMs can be trained on small graphs (e.g., 20) and generalize to larger graphs (e.g., 100). Thus, our solution is to develop techniques that can sample a subgraph from the large real-world graph and compensate for the information loss due to the sampling so that we train models on the small, sampled graph.
>
> ------
>
> Q2: Experimental setup in KG reasoning.
>
> A2: Thank you for the suggestion. We will clarify this in the paper. To clarify, it is true that in Sec 3.2, the input edge types and output edge types are different. In Sec 3.3., they are the same. However, the goal is similar: to learn rules that infer the edge type between some node pairs based on the input information.
>
> ------
>
> Q3: Training NLMs on KG datasets.
>
> A3: NLMs can not be applied to the KG datasets we have: they are real, large graphs, and it is unclear how to obtain “small graphs” from them. This is indeed the key contribution of this paper: how to find subgraphs to train hyper-graph models, compensate for the information loss during sampling, and reduce the memory cost in storing hyper-edges.
>
> ------
>
> Q4: Experiments around scalability analysis.
>
> A4: First, in this paper, the sparsity loss focuses on memory usage caused by storing hyperedges. This is substantial because this will make training and inference of hyper-graph reasoning models from impossible (training the NLM model without sparsity loss takes n^3 space) to possible (we only need n^2 space, see Figure 4). The justification of this loss term is shown in Figure 4: the n^3 curve is the scaling of our baseline (NLM), and our model uses an order of magnitude smaller memory.
>
> ------
>
> Q5: Ablation study on sparsification loss.
>
> A5: Per request, we have added an ablation study to compare our Hoyer-Square sparsification with L1 and L2 losses. The Hoyer-based loss is performing better than other approaches. The detailed results and analysis are in the appendix.
>
> ------
>
> Q6: Hyperparameter tau in the information sufficiency.
>
> A6: The reviewer raised a valuable point that the hyperparameter tau in the information sufficiency is crucial. When we have domain-specific knowledge, we can set this number to the length of the inference path required by the target relationship. For example, in the “grandparent” task, the length of the inference path is two (so we can set $\tau=2$). On real-world datasets (e.g., FB15k, NELL995), where we do not have domain-specific knowledge, we used $\tau=3$.
>
> ------
>
> Q7: Readability.
>
> A7: We agree that our work requires background knowledge of NLMs to comprehend fully. To address this, we have added the detailed implementation of operations of NLM in the appendix and a more comprehensible illustration and caption in Figure 2. Also, we detail the significance and necessity of these operations, such as EXPAND and REDUCE, in the appendix. We also updated the paper based on the reviewer’s questions.

---

### Official Review · Reviewer_emhK · 2021-11-02

**Correctness:** 3
**Technical Novelty And Significance:** 2
**Empirical Novelty And Significance:** 2
**Recommendation:** 6
**Confidence:** 3

**Main Review:**

**Strengths:**
1. Successfully able to scale NLM and demonstrate its effectiveness on 2 large-scale tasks.
2. Employ heuristics to encourage sparseness and efficient training: demonstrated through ablation as well.
3. Quantification of information content in a subgraph, and efficient subgraph sampling based on it.

**Weaknesses:**
My main concern is with the experiments in the paper. Please see the ‘summary of the review’ for the details.


**Summary Of The Paper:**

This paper presents a neural network, SpaLoc, which claims to improve the scalability of an existing architecture, Neural Logic Machines (NLM) (Dong et al).
The main idea of this paper is to exploit the sparseness of k-arity relationships that are represented as k dimensional dense tensors in Dong et al.  SpaLoc instead records the indices of non-zero entries and represents k-dim tensors as an (Index, Value) tuple (I,V) of 2 dimensional tensors, and redefines all the operations (Reduce, Expand and Permute) of NLM to suit the (I,V) representation.
It applies an auxiliary loss term to ensure sparseness of the intermediate as well as the final relationship learnt.
Finally, for efficient training (and inference), it relies on subgraph sampling techniques that are based on preserving the information content required to make a prediction. While training, they calibrate the prediction target based on the information lost due to subsampling.
They experiment on 2 tasks: synthetic family tree reasoning task, and  Knowledge Graph (KG) reasoning tasks. On synthetic data, they show that the original NLM fails to scale on very large families and a GNN based baseline (R-GCN) performs worse than SpaLoc. On inductive KG reasoning, it beats the existing methods.


**Summary Of The Review:**

It’s an incremental work, focused on scaling an existing method by converting a dense representation into a sparse representation.

Even though the theory is for generic arity, the experiments involve only binary relations, and hence many complex GNN based methods can be used in addition to R-GCN. For example, Recurrent Relational Networks (RRN)  (Palm etal 2018) that work very well for multi-hop reasoning and scale well across the size of the graph.

**Running time:**  For the sparse (I,V) representation, how are the three basic operations (Reduce, expand, permute) performed using fast tensor operations on GPU? In my understanding, one downside of sparsifying the tensors is: indexing now needs to be handled separately, and can’t be done using fast GPU tensor operations. All the three basic operations of NLM may require dictionary lookups done on CPU, instead of GPU. Hence, even though sparse tensors for very high arity may now fit in the GPU memory, performing basic NLM operations may require frequent interaction with CPU, and hence dramatically increasing the training / inference time.  Can you provide a comparison of the running times of different baselines with that of SpaLoc?

Experimental details are missing. For example, what is the maximum arity used in the intermediate layers for KG experiments? How many layers for both tasks?
Many baselines have not been cited or mentioned, except the acronym in the table (e.g. TACT in table 2).

**Clarification Questions:**

1. *Subgraph sampling:* The description for both sampling techniques (neighbor expansion and path sampler) focuses on 2 arity edges.
Specifically, in neighbor expansion, it appears that node set is expanded based on only binary relations? Are relations with higher arity used when expanding the node set? In path sampler, do you consider paths via edges of higher arity?

2. On page 6, under **Subgraph Sampling**:  "*During inference, usually our goal is to just infer the relationship between one pair of entities:*" In its original form, NLM would predict the relationship between all pairs of nodes as it works with dense representations. In SpaLoc, don’t we get all the non-zero entries, in the tensor of the desired arity, all at once in the forward pass?

3. *Section 2.3:*  How do you select hyperparameter \epsilon = 5%?

4. *Ablation study:* there is no ablation on ‘sparsification loss’. How important is it?


**Typos**

1. Page 2, Line 6: whether a predicate is true **for** some tuple of entities.

2. Section 2, 2nd line: why is V a  tuple?

3. Figure 2, Last row in the output of Permute and NN should be (3,4) and not (4,3)

4. References to Figures say Fig1b, Fig2b, but there are no parts a and b of the figures. For example, in the 1st line of section 2.2.

5. Section 2.2, page 4, in the 3rd line of the paragraph describing ‘EXPAND’, what is gF?

6. Section 2.2, page 4, equation in the description of ‘REDUCE’: if output of f is a vector, then shouldn’t it be a max pool (max across each dimension), instead of just max?

7. Page 4, last word: superscript should be (i,r) instead of (i,k)

8. Page 6, Information Sufficiency: You have defined S = (V_S, E_S), but you refer to S as H_S subsequently. Also, in the hyper-edge path, the subscript on the last node of the 2nd edge should be r_2 and not r_1.

9. Section 3.4: Ablation Study: it should be mentioned that all ablations are on synthetic family tree reasoning tasks.

---

> ### Author Response · Authors · 2021-11-23
> **Author Response to Reviewer emhK**
>
> Thank you for acknowledging the effectiveness of our work. We answer the questions below.
>
> ------
>
> Q1: Experiments for high-order relationships.
>
> A1: First, we would like to reiterate the importance of high-order edges: even if the input graph contains only binary relationships, incorporating high-order edges in intermediate features is still helpful. Please also see our general response.
>
> Per request, we have added new synthetic family tree reasoning tasks with input and output ternary predicates to address this point. Specifically, we add the family-of-three and the three-generations reasoning tasks. Below, we show that our model still achieves perfect accuracy on newly added tasks, while existing models except NLM cannot handle it.
>
> ------
>
> Q2: Running time comparison.
>
> A2: We have added a runtime comparison between NLM and SpaLoc in the appendix, showing that there is a constant level of optimization. We would like to reiterate that SpaLoc focuses on improving inference-time memory efficiency. We are the first model that can scale up hyper-graphs to real-world knowledge graph datasets.
>
> Figure 4 in the main paper shows the number of non-zero entries (NNZ) in the intermediate groundings of SpaLoc, which has a linear relationship with the computation complexity. Theoretically, the baseline NLM has $O(n^3)$ complexity on all three tasks, while our model is $O(n^2)$.  On small graphs, adding sparsity does not necessarily improve the running time on GPUs because of the overhead of handling sparse tensors due to the implementation. More efficient implementation of sparse operations is interesting future work.
>
> ------
>
> Q3: Experimental Details.
>
> A3: Thank you for the suggestion. We have added a new section in the appendix elaborating the experimental settings and details. We also revised the paper to cite all baselines.
>
> ------
>
> Q4: Subgraph sampling techniques applied to hypergraphs.
>
> A4: Yes, both the neighbor expansion and the path sampler can be applied to hypergraphs with any arity edges. We define two nodes on the hypergraph to be connected if and only if a hyperedge covers the two nodes. Therefore, the node set is expanded based on hyperedges of all parties.
>
> ------
>
> Q5: Single relation prediction in one forward pass?
>
> A5:  Yes! This is exactly how we train our models.
>
> In large-graph inference tasks, inferring the target relationship between all nodes is not feasible: there are $O(n^2)$ pairs, when $n > 10^4$ (e.g., FB15K), such dense representation will not fit into a single GPU. Thus, we infer all edges in the test dataset one-by-one. This is the setting for all baselines.
>
> ------
>
> Q6: The setting of the sparsity threshold.
>
> A6: We set $\epsilon=0.05$ in all our experiments. This value is chosen based on the validation accuracy of our model. In practice, we observe that any values between 0.01 and 0.1 do not significantly impact the performance of our method and the inference complexity.
>
> ------
>
> Q7: Ablation study on sparsification loss.
>
> A7: The usefulness of the sparsity loss is illustrated in Figure 4. The $n^3$ curve is the scaling of our baseline (NLM, which can also be interpreted as SpaLoc without the sparsity loss), and our model uses an order of magnitude smaller memory $O(n^2)$, which is the optimal efficiency, compared with human-defined rules. We have added an ablation study to compare our Hoyer-Square sparsification with L1 and L2 losses. The Hoyer-based loss is performing better than other approaches. The detailed results and analysis are in the appendix.

---

> > ### Comment · Reviewer_emhK · 2021-11-29
> > **Thanks for your responses**
> >
> > Thanks for your responses in the rebuttal. My concerns regarding actual training time and experiments using higher-order relationships have been addressed. I am happy to increase the rating to 6.

---

### Official Review · Reviewer_GMNn · 2021-11-02

**Correctness:** 3
**Technical Novelty And Significance:** 2
**Empirical Novelty And Significance:** 1
**Recommendation:** 3
**Confidence:** 4

**Main Review:**

As stated in the summary, this work proposes an NLM variant that seeks to solve the scalability issue. The proposed techniques are: using sparse representation, applying sparsity regularization, and using sub-graph sampling. These techniques do not pose significant changes to the original NLM model nor extend the model's capability in learning new classes of patterns. Furthermore, the techniques are specifically designed for the NLM, limiting the scope of this work. With that being said, I consider this work more or less incremental whose contribution is minor and limited. Therefore, I'm inclined to reject. My detailed comments are as follows:

The paper is difficult to read if the reader is unfamiliar with the NLM model
 - While this paper proposes a direct variant of NLM, it omits too many details on the original model. Personally, I'm not hindered by the writing as I'm familiar with NLM, but this is generally not true for most of the audiences with general KG background knowledge.
 - For example, section 2.2 introduces the EXPAND, REDUCE and PERMUTE operations but does not point out these operations are from NLM and why they are needed for KG reasoning.

How is the sparsity threshold epsilon set? Do authors run a grid search and set it to 5%?

What is N_train in section 3.2?

As the aim of SpaLoc is to address the scalability issue, the author should design the experiments around the scalability analysis. For example, besides Table 2, the author could provide the runtime comparison of each method. On the other hand, while I understand the authors choose to use the disjoint subset of FB15K and WN18 from (Teru et al., 2020) for inductive reasoning, the datasets are significantly smaller than the original benchmarks. For example, the largest dataset FB15K v5 only contains 5K nodes and 33K edges whereas the original FB15K contains 14K nodes and 272K edges. As the focus of this work is scalability rather than inductive capability, it is crucial to push the proposed method to its limit and evaluate it in a realistic setting.


**Summary Of The Paper:**

This paper proposes SpaLoc a Neural Logic Machine variant that improves the model scalability on the graph completion tasks. To do this, the authors 1) replace the original dense representation into sparse tensors; 2) apply a sparsity regularization on the original loss function; 3) propose a sub-sampling strategy that reduces the training complexity. The proposed method is evaluated with a set of differentiable ILP methods and GNN methods. Ablation studies are conducted to analyze the effectiveness of the proposed techniques.


**Summary Of The Review:**

In summary, SpaLoc is an incremental variant of the NLM model. The contribution is minor and limited to the specific NLM model. Therefore, I recommend rejection.

---

> ### Author Response · Authors · 2021-11-23
> **Author Response to Reviewer GMNn**
>
> Thank you for the insightful feedback. We answer specific concerns raised by the reviewer below.
>
> ------
>
> Q1: Readability for readers who are not familiar with NLMs.
>
> A1: Thank you for your suggestion. We have updated our manuscript to add a detailed explanation and pseudo-code implementation of Neural Logic Machines in the appendix. We have also updated the captions in Figure 2 for better illustration.
>
> ------
>
> Q2: The importance of EXPAND, REDUCE, and PERMUTE in KG reasoning.
>
> A2: With all three operations (and learnable MLP modules), NLMs and SpaLoc’s can realize first-order logic (FOL) rules (a detailed proof is in the NLM paper [1]). This involves useful transitive closures of the form p(x, y) <- q(x, z) ^ r(z, y), for example.
>
> ------
>
> Q3: The setting of the sparsity threshold.
>
> A3: We set $\epsilon=0.05$ in all our experiments. This value is chosen based on the validation accuracy of our model. In practice, we observe that any values between 0.01 and 0.1 do not significantly impact the performance of our method and the inference complexity. We have included this in the paper.
>
> ------
>
> Q4: Number of objects in training and testing (N_train).
>
> A4: In the original paper, we used two training and testing settings. Column 1 (train on $N=20$, test on $N=100$), Column 2 (train on $N=2000$, test on $N=100$). We fixed the test graph size to 100 because our baselines MemNN and NLM can not be applied (in testing) to larger graphs. We have clarified this in the paper.
>
> ------
>
> Q5: Experiments around scalability analysis.
>
> A5: Thank you for your suggestion. We added a scalability analysis in the appendix. In a nutshell, our method focuses on optimizing the space complexity of inference. The experimental results show that our method can reduce the space complexity from the original $O(n^3)$ complexity of NLM to the $O(n^2)$ complexity required to solve the has-sister and grandparent tasks. In terms of the inference time, there is a constant level of optimization.
>
> ------
>
> Q6: Experiments on larger knowledge graphs.
>
> A6: This paper focuses on the inductive link prediction task, as proposed in Teru et al. [2].
> In inductive link prediction, we need to split the original knowledge graph into three parts (training, validation, and test) with no overlapping nodes.
> We agree that we can test SpaLoc’s limit on larger newly built inductive link prediction datasets or just using the transductive setting. We apply SpaLoc to the complete real-world knowledge graphs compared with transductive learning methods. Experimental results are added in section 3.3. SpaLoc outperforms existing methods without leveraging any entity-level information learned from training, i.e., still using inductive learning.
>
> ------
>
> [1] Honghua Dong, Jiayuan Mao, Tian Lin, Chong Wang, Lihong Li, and Denny Zhou. Neural Logic Machines. In ICLR, 2019.
> [2] Komal K. Teru, Etienne Denis, and William L. Hamilton. Inductive Relation Prediction by Subgraph Reasoning. In ICML, 2020.

---

> > ### Comment · Reviewer_GMNn · 2021-11-29
> > **Thoughts after reading the rebuttal**
> >
> > Thank you for the rebuttal and the revised draft. The readability has definitely improved from the last draft. The new experiments on the transductive dataset have addressed my concerns on the scalability, though I would still prefer a runtime comparison side by side with Table 3.
> >
> > However, I'm still concerned about its novelty. The authors did not deny that this is a straightforward variant of NLM. The proposed techniques such as using sparse representation and subgraph sampling are straightforward. The proposed hypergraph viewpoint is interesting but it does not bring anything new to the table. For example: solving a new type of reasoning problem or learning an interpretable representation. That said, I'm keeping my score.

---

> > > ### Author Response · Authors · 2021-11-30
> > > **Response**
> > >
> > > Thank you for your thoughts. We would like to reiterate that our paper is novel and makes significant contributions over NLM. Specifically,
> > >
> > > 1. Our work is the first work showing that using hyperedges in real-world relational reasoning tasks is beneficial, even if the input graphs are binary. This was impossible for NLM or any other hypergraph neural networks because of their memory cost. We achieved two state-of-the-art results on real-world knowledge graph reasoning benchmarks (Inductive: Table 1 and Transductive: Table 3)
> > >
> > > 2. Subgraph sampling is highly nontrivial. Our proposal of using information sufficiency of the sub-sampled graphs to calibrate labels is completely new and crucial for performance. (Table 5)
> > >
> > > 3. Our work is the first work showing that by imposing specific sparsity losses, neural network-learned rules (SpaLoc) can be as efficient as human-defined first-order logic rules (Figure 4).
> > >
> > > We respectfully suggest that these points should be taken into consideration. Thanks.

---

### Official Review · Reviewer_joDw · 2021-11-03

**Correctness:** 3
**Technical Novelty And Significance:** 3
**Empirical Novelty And Significance:** 3
**Recommendation:** 6
**Confidence:** 4

**Main Review:**

strengths:
1) The study is based on a very nice observation that in logical reasoning we don't usually consider all entities but rather a local set of them, and the authors developed a novel method motivated by this observation that could improve efficiency by a lot.
2) the methods are sensible, from the sparse tensor representation to subgraph sampling to label calibration, all of which I can follow intuitively.
3) the experimental results looked very strong, especially on the synthetic datasets, which solved a large amount of instances that could not be solved by other methods given a fixed amount of memory budgets.

weakness:
The main problem I have for this paper is its presentation.  Specifically, the authors should improve on describing their methods and there is a lack of very important details in the experiments (even in the main part).
1) In figure 1(i), it's hard for me to tell why I+V should have a fixed length..  the first dimension of V decreases when the first dimension of I increases?
2) One of the most important claims in the paper is efficiency.  However, the authors are missing key details: for example, when the baseline methods hit OOM, what memory budgets did the authors put?  The authors also claim efficient inference time in various places, but not single empirical evidence is presented in the experiments section.
3) The new method achieves such amazing results on the synthetic datasets, but no single details are presented on the dataset was generated.

small comments:
1) the authors refer to fig 1b and fig 2b, but in both the figures, there are no sub-labeling
2) In fig 1, it's better add caption explaining what are I and V matrices

**Summary Of The Paper:**

Motivated by an observation that in logical reasoning, rules only apply locally and sparsely to a small set of entities, the authors propose Sparse and Local Neural Logic Machines (SpaLoc), a structured neural network for hypergraph reasoning. In particular, this paper has three contributions:

1) the authors developed a way to do sparse tensor representation to alleviate the computation burden in NLM
2) use hyperedge gates to enforce sparsity and impose sparsity-related loss
3) a novel strategy to do subgraph sampling that leverages a metric they invented called information sufficiency

Also showed positive experimental results compared to baseline methods and provided some ablation studies.

**Summary Of The Review:**

The paper is strong overall in terms of novelty of the methods and good main experimental results.  However, I would really like the authors to improve on the presentation of this paper.  In general, I lean towards accepting this paper, but in order for me to improve the score I need to see more details on their new methods are efficient in terms of time and memory and how the synthetic dataset is generated.

---

> ### Author Response · Authors · 2021-11-23
> **Author Response to Reviewer joDw**
>
> Thank you for acknowledging our work’s novelty and noting that our experimental results are strong.
>
> ------
>
> Q1: Demonstration in Figure 1.
>
> A1: Thank you for pointing out the potential confusion that I+V seems to have the fixed length in Figure 1. In SpaLoc, the dimension of I+V across different arities are not the same! We will update the figure and caption for better clarity.
>
> ------
>
> Q2: Experiments around scalability analysis: memory budget.
>
> A2: Thank you for your suggestion. We have included details about our memory setting in the paper. Specifically, for all methods, we have allocated 12GB of GPU memory. We added a scalability analysis in the appendix. In a nutshell, our method focuses on optimizing the space complexity of inference. The experimental results show that our method can reduce the space complexity from the original $O(n^3)$ complexity of NLM to the $O(n^2)$ complexity required to solve the has-sister and grandparent task. In terms of the inference time, there is a constant level of optimization.
>
> ------
>
> Q3: Experiments around scalability analysis: running time.
>
> A3: Figure 4 in the main paper shows the number of non-zero entries (NNZ) in the intermediate groundings of SpaLoc, which has a linear relationship with the computation complexity. Theoretically, the baseline NLM has $O(n^3)$ complexity on all three tasks, while our model is $O(n^2)$. On small graphs, adding sparsity does not necessarily improve the running time on GPUs because of the overhead of handling sparse tensors due to the implementation.
>
> Meanwhile, we would like to reiterate that SpaLoc focuses on improving the inference-time memory efficiency, as illustrated in A2. We are the first model that can scale up hyper-graphs to real-world knowledge graph datasets. More efficient implementation of sparse operations is interesting future work.
>
> ------
>
> Q4: Details about synthetic dataset generation
>
> A4: Thank you for the suggestion. We have added formal details of how we generated the family tree reasoning tasks in the appendix. They followed the previous work Neural Logic Machines [1].
>
> ------
>
> Q5: Small comments.
>
> A5: Thank you for your suggestions on figure details. We have updated our manuscript to incorporate them.
>
> ------
>
> [1] Honghua Dong, Jiayuan Mao, Tian Lin, Chong Wang, Lihong Li, and Denny Zhou. Neural Logic Machines. In ICLR, 2019.

---

> > ### Comment · Reviewer_joDw · 2021-11-30
> > **Thanks for your rebuttal**
> >
> > I have changed my score to 6 to accept this paper. I think the additional details provided in the revision is helpful in demonstrating the effectiveness of their approach and the results are very promising.

---

### Official Review · Reviewer_Rg6r · 2021-11-04

**Correctness:** 3
**Technical Novelty And Significance:** 3
**Empirical Novelty And Significance:** 3
**Recommendation:** 6
**Confidence:** 3

**Main Review:**

Strengths:

1. Developing models for hyperrelation is interesting and under explored. The main contribution of paper is to make neural logic machines practical and scalable which is important
2. The method is well-motivated. For example, the use of Hoyer measure and subgraph sampling and training with information sufficiency is interesting and makes sense
3. They achieve good results on a synthetic and few real world KG benchmarks

Questions for authors:

1. The current experiments (relation extraction) and family tree are all finding a binary relation between a pair of entities. Since the method is developed for hyperedges, why is there not an experiment where the goal is to infer relations between more than a pair of entities?
2. How is the arity set? For example, for the family tree experiemnts, how did you come up with value 3? Is it set as a hyperparameter.
3. In table 1, why does not memory networks scaled to 2000 objects? Is this an implementation issue?
4. In table 1, what does tested on 100 objects mean? I thought the number of objects are 20 and 2000?
5. I think the writing of the paper needs to significantly improved. For example, Figure 1 is very hard to understand for a reader who is not familiar with NLMs. Also readability of Figure 2 can be improved by adding proper captions as that is the main figure explaining the mechanism of the model.

**Summary Of The Paper:**

This paper proposes extensions of Neural Logic Machines (NLM, Dong et al 2019) to make them more scalable. NLMs operate on dense hypergraph representations. Since the space requirement is exponential in terms of number of entities, NLMs become quickly intractable as the number of entities increase. To remedy this, this paper proposes a sparse tensor based representation for hyoeredge. The expand, reduce and permute operations are also defined for sparse tensors. To achieve sparsity in training, they include a sparse regularization objective (based on Hoyer measure). Lastly, they also employ a subgraph sampling technique based on information sufficiency.

On synthetic family tree relationships, the proposed model solves the dataset and is the only model that scales to 2000 entities. Moreover they achieve the optimal sparsity (quadratic).

**Summary Of The Review:**

This is definitely interesting work and I think efficiently modeling hyper-edges is important. However, currently, the paper is very dense to read and needs to be improved. I would also like to see an experiment on inferring relations that exists between more than a pair of entities.

=====Update 11/26=======

Thank you for the rebuttal. I am keeping my score to weak accept.

---

> ### Author Response · Authors · 2021-11-23
> **Author Response to Reviewer Rg6r**
>
> Thank you for acknowledging our work’s significance and noting that our experimental results are promising. We address the concerns below.
>
> ------
>
> Q1: Experiments for high-order relationships.
>
> A1: First, we would like to reiterate the importance of high-order edges: even if the input graph contains only binary relationships, incorporating high-order edges in intermediate features is still helpful. See also our general response.
>
> Per request, we add two new synthetic family tree reasoning tasks with input and output ternary predicates to address this point: family-of-three and three-generations reasoning. Our model still achieves perfect accuracy on both tasks, while existing models except NLM cannot handle it.
>
> ------
>
> Q2: Arity setting in SpaLoc.
>
> A2: Across all experiments in this paper, the maximum arity of intermediate predicates (i.e., the “breadth”) is set to 3 as a hyperparameter. Conceptually, this allows SpaLoc to realize all first-order logic (FOL) formulas with at most three variables, such as a “transitive relation rule.”
>
> ------
>
> Q3: Scalability of MemNN to 2000 objects.
>
> A3: In the paper, we followed the settings in Neural Logic Machines [1] to encode a binary relationship using $O(n^2)$ entries (a dense representation) as the input to the Memory Network. Thus, it failed to scale up to 2000 objects.
>
> ------
>
> Q4: Number of objects in training and testing.
>
> A4: In the original paper, we used two training and testing settings. Column 1 (train on $N=20$, test on $N=100$), Column 2 (train on $N=2000$, test on $N=100$). We fixed the test graph size to 100 because our baselines MemNN and NLM can not be applied (in testing) to larger graphs. We have clarified this in the paper.
>
> ------
>
> Q5: Readability for readers who are not familiar with NLMs.
>
> A5: Thank you for your suggestion. We have updated our manuscript to add a detailed explanation and pseudo-code implementation of NLM in the appendix. We have also updated the captions in Figure 2 for better illustration.
>
> ------
>
> [1] Honghua Dong, Jiayuan Mao, Tian Lin, Chong Wang, Lihong Li, and Denny Zhou. Neural Logic Machines. In ICLR, 2019.

---

### Author Response · Authors · 2021-11-23
**Summary of the Paper Revision**

We thank all reviewers again for their constructive reviews. We have updated our manuscript accordingly. Specifically:

- New result: We have added new results on the knowledge-graph transductive reasoning tasks, which consider larger real-world graphs than the existing ones. We outperforms all baselines and achieved a new state-of-the-art.

- New result: We have added an ablation study comparing different sparsification losses. The Hoyer-Square measure outperforms L1 and L2.

- New result: We have added an ablation study measuring the actual GPU memory usage of our model. We see a nearly $O(n^2)$ memory usage (c.f., the original NLM is $O(n^3)$). This is consistent with our Figure 4 measure of non-zero elements (NNZ).

- Implementation detail: We have added new sections in the appendix about the experiment settings.

- Implementation detail: We have added new sections in the appendix about the calculation of information efficiency and the implementation of our graph samplers.

- Readability: We have updated our manuscript to add a detailed explanation and pseudo-code implementation of NLM in the appendix. We have also updated the captions in Figure 2 for better illustration.

---

### Author Response · Authors · 2021-11-23
**General Response to Reviewers**

We thank all reviewers for their insightful reviews and helpful, constructive comments. In our general response, we would like to clarify and reiterate the motivation and the contribution of this paper.

In general, we want to learn models that can predict relationships among a set of entities. Consider simply the prediction of kinship relationships. Even if all input and output relations are binary edges (father, mother, uncle, etc.), it is still important to consider high-order relationships among nodes. For example, to derive the “grandparent (gp)” relationship, we need to consider all three-tuples $(x, y, z)$ because $gp(x, z) \leftarrow parent(x, y) \land parent(y, z)$. In general, following Barceló et al.[1], realizing a logical formula with k variables needs k-ary hyper-edges.

We have seen the success of hyper-graph relational neural networks such as Hyper-graph GNNs [2] and Neural Logic Machines [NLMs; 3]: they gain expressiveness by incorporating hyper-edges, but they cannot scale up to real-world tasks, most notably due to the impractical memory usage used for storing all k-ary relations (recall that an order-k hypergraph requires up to $O(n^k)$ space).

Our motivation is to use the natural sparsity and locality of these problems to make the model more efficient, enabling applications that were impossible before this work. The technical methods for taking advantage of sparsity and locality are highly nontrivial and should not be considered incremental.

- First, sparse methods for standard graph neural-network implementations only consider edges in the input graph, but this does not generalize to hyper-edges. As mentioned, to infer the grandparent relationship between two nodes x and z, we need to consider all tuples $(x, y, z)$. As humans, we know it’s sufficient to consider tuples that satisfy $parent(x, y)$ and $parent(y, z)$, but machines need to learn how to focus on a subset of all $(x, y, z)$’s from data. In this paper, we employ a training-time regularization to encourage representation sparsity.

- Second, we consider using sub-sampled graphs for training and inference, which is an underexplored domain for inductive reasoning tasks. We show that different sampling strategies can make a huge difference because subsampling is always lossy: the information in a subsampled graph may be insufficient to infer the target relation. In this paper, we have investigated different sampling strategies and developed a new label calibration strategy to compensate for the information loss due to sampling.

Our model addresses both challenges. Our experiments on synthetic datasets show that our sparsity regulation encourages the model to consider only the minimal set of edges (the order is quadratic instead of cubic) without sacrificing performance. More importantly, to the best of knowledge, our paper is the first evidence that hyper-edge relationships are essential in inferring relations in real-world knowledge graphs: such as FB15k, NELL995, etc. Moreover, our model is the first hyper-graph NN implementation that can scale up to the real-world knowledge-graph inference challenges. Our model has out-performed all state-of-the-art methods that only consider binary edges.

[1] Pablo Barceló, Egor V Kostylev, Mikael Monet, Jorge P ́erez, Juan Reutter, and Juan Pablo Silva. The logical expressiveness of graph neural networks. In ICLR, 2020.

[2] Christopher Morris, Martin Ritzert, Matthias Fey, William L. Hamilton, Jan Eric Lenssen, Gaurav Rattan, and Martin Grohe. Weisfeiler and Leman Go Neural: Higher-order Graph Neural Networks. In AAAI, 2019.

[3] Honghua Dong, Jiayuan Mao, Tian Lin, Chong Wang, Lihong Li, Denny Zhou. Neural Logic Machines. In ICLR, 2019.

---

### Author Response · Authors · 2021-11-29
**Looking Forward to Reviewers' Feedback**

Dear reviewers,

Thank you again for your reviews, comments, and suggestions. As the end of the discussion period is approaching, we want to be sure we have addressed all of your concerns. Please let us know if you have any additional comments.

---

### Decision · Program_Chairs · 2022-01-20

**Decision:**

Reject

**Comment:**

This paper has conflicting reviews with no strong advocate.  One of the positive reviewers states the caveat that paper is "very dense to read and needs to be improved".  Having looked at the paper myself I would agree with this criticism.  One of the negative reviewers states that the paper gives "an incremental variant of the NLM model".  I am less confident in this judgement.  However, I find the density of the paper and the use of synthetic data to be significant drawbacks. With the lack of any real champions for the paper I do not see a path to acceptance.